# GUIDING VLM AGENTS WITH PROCESS REWARDS AT INFERENCE TIME FOR GUI NAVIGATION

## ABSTRACT

Recent advancements in visual language models (VLMs) have notably enhanced their capabilities in handling complex Graphical User Interface (GUI) interaction tasks. Despite these improvements, current frameworks often struggle to generate correct actions in challenging GUI environments. State-of-the-art commercial VLMs are black-boxes, and fine-tuning open-source VLMs for GUI tasks requires significant resources. Additionally, existing trajectory-level evaluation and refinement techniques frequently fall short due to delayed feedback and local optimization issues. To address these challenges, we propose an approach that guides VLM agents with process supervision by a reward model during GUI navigation and control at inference time. This guidance allows the VLM agent to optimize actions at each inference step, thereby improving performance in both static and dynamic environments. In particular, our method demonstrates significant performance gains in the GUI navigation task setting, achieving a 5% improvement in action accuracy for static environments and a around 15% increase in task success rate in dynamic environments. With further integration of trajectory reflection and retry mechanisms, we also demonstrate even greater enhancement in task success.

## 1 INTRODUCTION

Recent advances in VLMs have significantly enhanced their capabilities in understanding, reasoning, and generalizing, enabling them to handle complex real-world GUI interaction tasks (Hong et al., 2024b; You et al., 2024; Cheng et al., 2024). For instance, given an instruction like "How do I get to the nearest Walmart?", a VLM agent is expected to navigate to the Google Maps application, search for Walmart locations in the vicinity, and select the nearest one to initiate route navigation. These advancements greatly improve the accessibility and efficiency of GUI interaction tasks.

However, even state-of-the-art visual language models (VLMs) like GPT-4V (OpenAI, 2023), Gemini 1.5 Pro (Reid et al., 2024) and others, as well as interaction agent frameworks like Yan et al. (2023); Wang et al. (2024); Zhang et al. (2024), still struggle to generate correct actions when completing GUI tasks such as VisualWebarena (Koh et al., 2024), OSWorld (Xie et al., 2024) and others. These commercial VLMs are typically black-box models, making them **inaccessible for tuning**, and further **fine-tuning open-source VLMs for GUI tasks remains resource-intensive**. Additionally, Pan et al. (2024) introduce a technique where GPT-4V serves as an evaluator to assess task success and provide reflection for retrying in case of failure, which can enhance the performance of agents in GUI navigation and control. However, such evaluation and refinement methods at the end of a trajectory will lead to **local optimization deficiency and delayed feedback**. Evaluating only at the end of the trajectory can result in insufficient optimization of individual actions, overlooking the refinement needed at each step. In GUI tasks, where each step impacts the final outcome, neglecting step-by-step optimization may degrade overall performance. Moreover, trajectory-level evaluation delays error correction, increasing both computational and time costs. Meanwhile, Bai et al. (2024) propose DigiRL, improving task performance in dynamic environments by combining Advantage-Weighted Regression with online reinforcement learning (RL) and an automatic curriculum mechanism. Such RL methods can lead to **high computational and time costs, along with a complex training process** that requires extensive online interaction data. Moreover, the training of RL algorithms is often unstable due to the sparsity and uncertainty of feedback, as well as the inherent trade-off between exploration and exploitation. These factors contribute to the high training cost and prolonged training time, particularly in dynamic and complex environments.

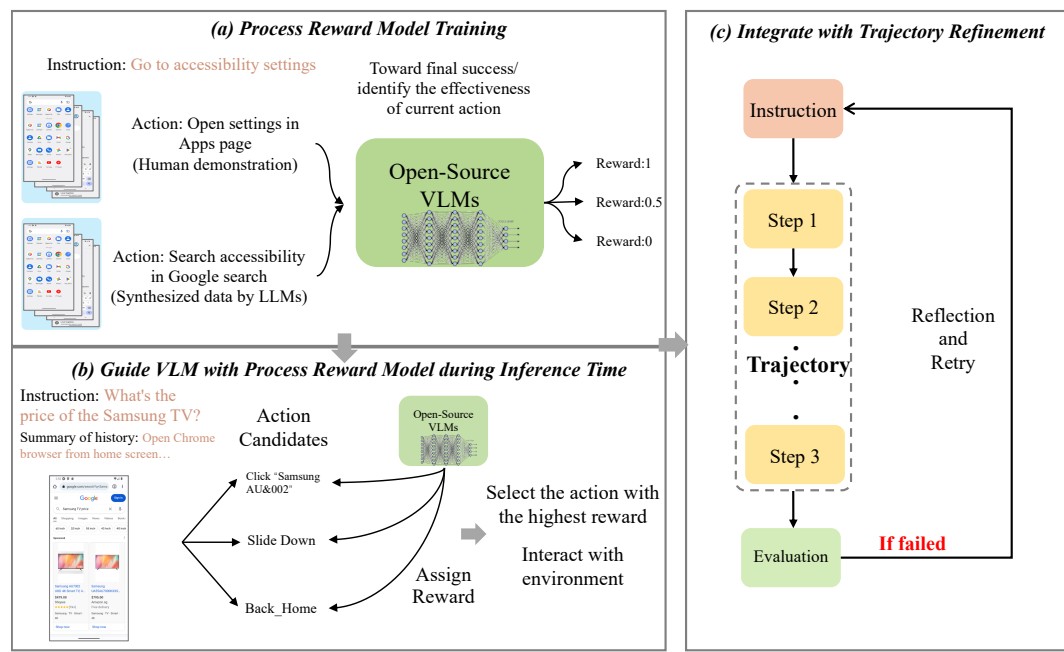

Figure 1: Overview of GuidNav.

To address these limitations, we propose **GuidNav**, guiding the VLM agent with a process reward model during interaction inference in GUI navigation and control tasks. Our empirical findings, along with OpenAI's o1 results (OpenAI, 2024), show that increasing effort during inference can significantly improve performance. Furthermore, there are strong reasons to favor process supervision through a reward model. It offers more precise feedback by identifying the exact step where an error occurs. This approach directly rewards models for following a path to success. In contrast, models guided by trajectory-based outcome supervision often take inefficient actions, deviating from the correct path and requiring additional effort to correct. Process-based rewards can help reduce these deviations, leading to a more efficient action trajectory. Guiding the VLM agent with this process reward model enables the agent to learn which actions are effective for achieving the given goal within the GUI task environment. As illustrated in Figure 1, to achieve this guidance, we first train a process reward model based on a limited amount of human demonstrations and synthesized data generated by the VLM. This process reward model learns the feedback signal from GUI data, guiding the VLM agent during GUI navigation inference to ensure it executes optimal actions. By providing process reward feedback at each inference step, the VLM agent can more accurately adjust its behavior. This fine-grained optimization enhances the success rate of tasks, especially in complex GUI environments where the correct execution of each step is crucial. Furthermore, unlike delayed feedback in trajectory-level evaluation, step-level process rewards enable the model to learn and adapt to environmental changes in real-time, preventing the accumulation of errors caused by delayed feedback. Additionally, like the demonstration in Figure 1 (c), our process reward model can also been integrated into outcome supervision pipline to further improve action generation and selection.

We evaluate GuidNav in both static and dynamic settings within Android-in-the-Wild (AitW) (Rawles et al., 2024b), measuring action accuracy at each step based on existing annotations and overall task success through human evaluation. The experimental results demonstrate that our method improves GPT-4o by around 5% in action accuracy of static environments and about 15% in task success rate within dynamic environments. With trajectory reflection and retry mechanisms, the success rate can reach a peak of 75.9%. In summary, our contributions are primarily in the following three areas:

- We introduce GuidNav, an approach that guides VLM agents for action decision during GUI interactions through a process reward model.
- Our approach can be easily integrated into trajectory-level refinement to further strengthen the performance.
- We show that our method can enhance VLM agents in both static and dynamic settings.

## 2 RELATED WORK

**GUI Navigation Agents and Benchmarks** Previous GUI navigation agents primarily focused on text-based prompts describing the environment, such as the HTML code, Document Object Model (DOM), or accessibility trees. However, current research leverages both screenshots and text instructions to navigate interfaces more akin to human-environment interactions. For instance, Auto-UI (Zhan & Zhang, 2023) utilizes GUI data to tune the language and projection modules, enabling interaction in multimodal GUI environments without the need for environment parsing or application-dependent API access. AppAgent (Yang et al., 2023b) employs the vision capabilities of large language models to operate smartphone applications in a human-like manner. Mobile-Agent-v2 (Wang et al., 2024) presents a multi-agent architecture for assisting mobile device operations. OS-Copilot (Wu et al., 2024) accelerates the development of computer agents on Linux and MacOS by providing a universal interaction interface. MM-Navigator (Yan et al., 2023) generates executable actions based on the screen image, text instructions, and interaction history. CogAgent (Hong et al., 2024b) leverages extensive GUI grounding data to further train the VLM for enhanced interaction. Additionally, several works focus on visual interaction tasks across app, web, and OS environments. AitW (Rawles et al., 2024b) and Weblinx (Lù et al., 2024) use human demonstrations to evaluate the accuracy of proposed actions. Osworld (Xie et al., 2024), AgentStudio (Zheng et al., 2024), AndroidWorld (Rawles et al., 2024a), and Visualwebarena (Koh et al., 2024) provide simulation environments for executing arbitrary agent trajectories in various domains and tasks. While these are not yet perfect, they serve as suitable platforms for assessing agents' capabilities.

**Evaluation by Reward and Reinforcement Learning Methods** Apart from agent framework, some researchers use reinforcement learning and reward models to enhance VLM agents further. Pan et al. (2024) introduce an Autonomous Evaluator for agent behavior, refining the agent's ability through reflection or fine-tuning based on filtered behavior cloning data. Bai et al. (2024) improve task performance in dynamic environments by combining Advantage-Weighted Regression with online reinforcement learning and an automatic curriculum mechanism. Zhai et al. (2024) employ a reinforcement learning method with a game-rule-based reward to strengthen VLM-powered agents in Gym Cards and ALFWorld. Fereidouni & Siddique (2024) utilize a two-stage learning process: Supervised Learning, where human demonstration data maps state to action, and Unsupervised Learning, where the PPO algorithm fine-tunes by optimizing policy gradients. A language model calculates action probabilities based on user goals and observations. Compared to process reward models, these methods often struggle to provide fine-grained step-level feedback and tend to incur significantly higher training costs.

**Discussion** Unlike methods like DigiRL (Bai et al., 2024) and Autonomous Evaluator (Pan et al., 2024) that rely on trajectory-level feedback or high computational training costs, GuidNav provides step-level rewards, enabling immediate optimization and reducing computational costs. This approach offers more efficient and precise action refinement, improving task performance in both static and dynamic environments.

## 3 METHOD

The primary task of GUI navigation involves enabling the VLM agent to interpret task instructions and interact with GUI screenshots to achieve a desired goal. Formally, let $x$ represent the task instruction, and let $S_t = \{s_1, s_2, \ldots, s_t\}$ represent the sequence of GUI states observed at different time steps $t$. The VLM agent must generate and select actions $A_t = \{a_1, a_2, \ldots, a_t\}$, where $a_t$ is the action taken at time step $t$, that modify the GUI state from $s_t$ to $s_{t+1}$. The overall goal is to generate and determine an action sequence $A$ that leads the GUI environment from the initial state $s_1$ to a final state $s_T$ that satisfies the task's objective. The core challenge is to enable the VLM agent to generate and determine actions $a_t$ at each time step that are most aligned with the task's goal $g$, while minimizing errors or irrelevant operations. The method, GuidNav, we propose consists of a two-stage process aimed at training the process reward model and guide VLMs agent for efficient task execution in GUI task environments at each time step $t$. The stages are as follows:

### 3.1 REWARD MODEL TRAINING

As shown in Figure 1 (a) and task definition, given a user instruction $x$, the historical states $S_{t-1} = \{s_1, s_2, \ldots, s_{t-1}\}$ and corresponding actions $A_{t-1} = \{a_1, a_2, \ldots, a_{t-1}\}$, as well as the current state $s_t$ and an action candidates $a_t$, the VLM as process reward model $\mathbb{R}$ assigns a reward $r$ to a given

action candidate $a_t$ in the context of the user instruction $x$ and current $(S_{t-1}, A_{t-1})$. The training data for this process reward model can be obtained through two primary sources:

1. **Human Demonstrations**: This involves collecting data from human experts who interact with the environment, providing a trajectory of states, actions, and corresponding rewards, i.e., $\{x^{(i)}, (S^{(i)}, A^{(i)}, R^{(i)})\}_{i=1}^{N}$, where $R^{(i)}$ represents the reward sequence for trajectory $i$. The reward value $r_t$ at time step $t$ of a trajectory is typically set to 1, as each action candidate is carefully selected by human experts and is assumed to be correct by default.

2. **Self-Playing via VLMs**: VLMs simulate interactions with the environment, generating synthetic trajectories of states and actions $\{x^{(j)}, (S^{(j)}, A^{(j)}, \tilde{R}^{(j)})\}_{j=1}^{M}$, where $\tilde{R}^{(j)}$ denotes the synthesized reward sequence. The reward $r_t$ is assigned based on the effectiveness of the VLM - generated action candidate $a_t$ in achieving the user instruction $x^{(j)}$ and the task's goals, rather than merely its similarity to human demonstrations. Details regarding the value assignment of $r_t$ are provided in Appendix 8.1.

To reduce the input length, thus mitigating potential degradation in performance due to excessively long inputs, at each time step, the VLM is employed to perform multimodal self-summarization based on the prompt $P$ (elaborated in Appendix 8.3), which converts the historical state and actions $(S, A)$ into a concise history in the form of natural language:

$$h_t = \textbf{VLM}((S_{t-1}, A_{t-1}), P)) \tag{1}$$

Then, the reward $r_t$ guided by process reward model $\mathbb{R}$ and assigned to action candidate $a_t$ can be represented as:

$$r_t = \mathbb{R}(x, h_t, s_t, a_t) \tag{2}$$

**Training Objective:** The process reward model $\mathbb{R}$ is trained to minimize the difference between the predicted rewards and the annotated rewards by minimizing a loss function. Specifically, the objective is to minimize the Mean Squared Error across all trajectories and their respective time steps between the predicted reward $r_{t,\text{pred}}$ for each action candidate and the annotated reward $r_{t,\text{anno}}$:

$$\mathcal{L}(\theta) = \frac{1}{\sum_{i=1}^{N} T^{(i)}} \sum_{i=1}^{N} \sum_{t=1}^{T^{(i)}} \left( r_{t,\text{pred}}^{(i)} - r_{t,\text{anno}}^{(i)} \right)^2 \tag{3}$$

Here, $N$ represents the total number of trajectories, and $T^{(i)}$ is the number of time steps in trajectory $i$. The term $r_t^{(i)}$ refers to the predicted reward at time step $t$ in the $i$-th trajectory, and $r_{t,\text{true}}^{(i)}$ is the corresponding annotated reward.

## 3.2 GUIDE VLMs WITH A PROCESS REWARD MODEL

We demonstrate the process in Figure 1 (b), as the aforementioned reward model training.

**Action Generation** We follow the similar strategy for VLM interaction inference in GUI tasks. Given a user instruction $x$, the historical states $S_{t-1} = \{s_1, s_2, \ldots, s_{t-1}\}$ and corresponding actions $A_{t-1} = \{a_1, a_2, \ldots, a_{t-1}\}$, the VLM serves as policy model $\mathbb{P}$ will first summarize the previous states and actions to obtain a concise history summary $h_t$. Thus, the user instruction $x$, history summary $h_t$, current time step state $s_t$ and corresponding prompt $P_{\text{inference}}$ (Appendix 8.3) will be used as input of VLM to generate $k$ possible actions $\mathcal{A}_t = a_t^1, a_t^2, \ldots, a_t^k$. This can be formulated as:

$$\mathcal{A}_t = \mathbb{P}(x, h_t, s_t, P_{\text{inference}}) \tag{4}$$

**Reward Assignment** According to Equation 2, the reward model assigns a scalar reward $r_t^k$ for each action candidate $a_t^k$ based on its alignment with the task. The reward $r_t^k$ is calculated as:

$$r_t^k = \mathbb{R}(x, h_t, a_t^k, s_t) \tag{5}$$

**Action Selection** The VLM selects the action $a^*$ with the highest reward $r^*$:

$$a_t^* = \arg\max_{a_t^k} r_t^k \tag{6}$$

This selected action $a^*$ is then executed to interact with the environment. The process is iteratively refined to improve the alignment of the VLM's actions with the desired outcomes, ensuring that the actions taken are those most likely to achieve the user's objective.

### 3.3 TRAJECTORY REFINEMENT AND EVALUATION

The process reward model can also been integrated with the refinement of the trajectories generated by the VLM to further enhance the performance, as depicted in Figure 1 (c).

**Trajectory Formation**    Once an action $a$ is selected and executed, it becomes part of the trajectory $T = (s_1, a_1), (s_2, a_2), \ldots, (s_t, a_t)$. The trajectory represents the sequence of state-action pairs leading from the initial state toward the task objective.

**Evaluation and Reflection**    At the end of each trajectory, the VLM evaluates the success of the trajectory in achieving the desired outcome. If the trajectory fails to meet the desired criteria, the VLM reflects on the reasons for the failure, generating a "reflection thought" that encapsulates the lessons learned from the unsuccessful attempt. This reflective thought is then incorporated into the retry process, informing the next iteration.

**Reflection and Retry**    The reflective thought generated by the VLM becomes part of the input for the next attempt. The VLM uses this enriched input, including the original instruction and the new reflective thought, to generate a new trajectory. This iterative process of reflection and retry continues until the VLM successfully achieves the task objective. Once a successful trajectory is identified, it is confirmed, and the process is completed.

## 4 EXPERIMENT

### 4.1 BASELINES

**Direct Prompting (DP)** involves directly prompting a Visual Language Model (VLM) to generate an action based on the instruction query, the current screenshot, and a summary of the previous state.

**TopK** is a technique where the model generates the top $k$ most probable actions (Xiong et al., 2023; Tian et al., 2023). In this procedure, while the model generates $k$ actions (with $k$ set to 3 in our work), we simplify the process by always selecting the most probable one (the first action in the list). This ensures that the model still considers multiple possibilities but prioritizes the highest-probability action for execution.

**Reflection** (Shinn et al., 2024) is a framework that improves LLMs' decision-making abilities in various tasks by using linguistic feedback and episodic memory, achieving significant performance gains in environments.

**Autonomous Refinement (AR)** (Pan et al., 2024) leverages the Reflexion technique (Shinn et al., 2024), an agent first attempts a task, and an external evaluator is used to judge whether its attempt was successful or not. If it is judged as unsuccessful, the agent will be prompted to reflect on the failure and retry. Here, we utilize GPT-4o as the external evaluator.

**DigiRL** (Bai et al., 2024) is an autonomous RL approach, for training in-the-wild device control agents through fine-tuning a pre-trained VLM in two stages: offline RL to initialize the model, followed by offline-to-online RL.

### 4.2 DATASET

**Android-in-the-Wild** (AitW) (Rawles et al., 2024b) is a large-scale dataset for Android device control, comprising 715,142 human demonstrations across 30,378 unique instructions. These instructions are divided into four subsets: General, WebShopping, GoogleApps, and Installation. We leverage ground-truth data and self-play data from 300 tasks in each subset to form the training set for the process reward model. Following the approach outlined in Yan et al. (2023), we randomly select 300 tasks from the AitW test set to evaluate action accuracy in a static environment. To ensure balanced representation, each subset contributes 75 tasks. For dynamic environment evaluation, we sample 120 tasks (30 from each subset) as instruction queries in a simulated setting. We also attach the comprehensive action space of AitW in Appendix 8.4.

## 4.3 SETUP

We utilize GPT-4o as the VLM policy model (set the tempature as 0.8) and leverage CogVLM2 (Hong et al., 2024a) as process reward model. Additionally, we leverage Set-of-Mark (SoM) method (Yang et al., 2023a) to enable the communication between VLM and screen. More details about SoM setting can be found in Appendix 8.2. In terms of static evaluation, we follow the previous settings (Rawles et al., 2024b; Yan et al., 2023) and compute the screen-wise partial action matching score as the main evaluation metric, defined as the number of correct actions divided by the episode length, then this score is averaged over all tested episodes. As for dynamic assessment, we call for 2 annotators to measure final success for each task.

| | General | Google_apps | Install | Web_shopping | Average |
|---|---|---|---|---|---|
| Topk w/ Oracle Eval | 55.8 | 49.5 | 54.4 | 57.2 | 53.7 |
| DP | 30.3 | 39.1 | 36.2 | 34.4 | 34.3 |
| TopK | 31.0 | 35.8 | 34.4 | 36.9 | 34.0 |
| Reflection (Shinn et al., 2024) | 31.2 | 37.9 | 32.6 | 30.0 | 32.9 |
| GuidNav | 35.5 | 41.4 | 40.9 | 38.5 | 38.9 |
| GuidNav Pass@N | **43.4** | **48.4** | **48.8** | **42.3** | **46.8** |

Table 1: Performance comparison of approaches in *static assessment* across four AitW tasks. Topk w/ Oracle Eval uses an oracle to select the best action from the top-K candidates. Pass@N, with N set to 3, calculates the action accuracy across multiple attempts, counting how many outcomes are correct in 3 trials.

| | General | Google_apps | Install | Web_shopping | Average |
|---|---|---|---|---|---|
| DP | 48.3 | 51.7 | 29.4 | 17.4 | 38.8 |
| Topk | 40.0 | 45.5 | 41.2 | 25.0 | 43.3 |
| AR (Pan et al., 2024) | 59.3 | 50.0 | 41.9 | 17.6 | 42.2 |
| DigiRL (Bai et al., 2024) | 56.3 | - | - | 32.7 | - |
| GuidNav | 58.6 | 64.5 | 47.8 | 35.3 | 54.0 |
| Integration | **80.0** | **91.3** | **64.7** | **53.8** | **75.9** |

Table 2: Performance comparison of approaches in *dynamic assessment* across four AitW tasks. Integration refers to the method where we combine the process reward model with the AR approach (3 retries), as detailed in Section 3.3.

## 5 PERFORMANCE

### 5.1 EXPERIMENTAL RESULTS IN AITW

**Static Evaluation** As shown in Table 1, compared to DP, our method achieves an average improvement of 4.6%, particularly in the 'General' domain tasks where we see a 5.2% gain. Simply applying the TopK method does not yield the same benefit. However, when we incorporate a reward model to identify the most likely action, the improvement becomes even more substantial. Furthermore, among the top $k$ possible actions, selecting the action based on oracle evaluation (Topk w/ Oracle Eval) reveals a high upper bound. This indicates that while VLMs can generate a potentially correct action, the correct one is often not their first choice. By applying our method multiple times (GuidNav Pass@N), we can significantly enhance its overall performance.

**Dynamic Assessment** In the dynamic environments of table 2, we use human evaluation to assess task success rates, offering a more realistic evaluation aligned with real-world scenarios. Our GuidNav outperforms both DP and AR, achieving overall improvements of approximately 15.2% and 11.8%, respectively. Even when compared to the DigiRL method, which includes further tuning via reinforcement learning, GuidNav maintains superiority. Additionally, our method provides process-level supervision, whereas the AR approach evaluates the final outcome and offers insights for retries. These two methods can be naturally integrated (the results of 'Integration'), enabling us to achieve a higher success rate with a maximum of three retries.

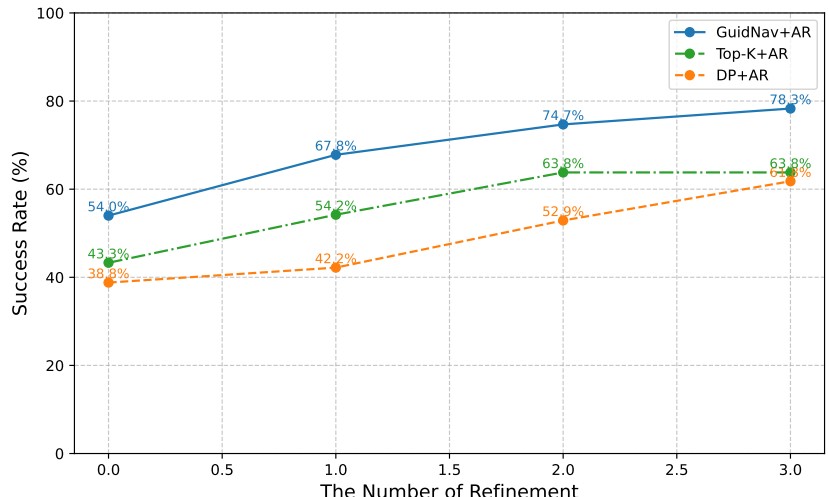

Figure 2: The performance curve across different trial numbers shows the impact of refinement techniques. 'DP+AR' represents the combination of direct prompting for action at each step, followed by AR at the end of each trajectory trial. 'GuidNav+AR' integrates TopK action selection guided by a reward model, with AR applied at the end of each trajectory trial. 'TopK+AP' refers to TopK method integrated with AR.

## 5.2 SELF-IMPROVEMENT IN OPEN SOURCE VLM POLICY MODEL

Based on the guidance provided by the process reward model, the policy model can be enhanced during inference. Additionally, the model's intrinsic abilities are strengthened as a result of the process reward model's guidance through successful trajectory fine-tuning. Table 3 demonstrates the benefit of increasing successful trajectories. With 300 and 800 trajectories data, the model's performance can be further improved.

| Method | General | Google_Apps | Install | Web_shopping | Average |
|---|---|---|---|---|---|
| VLM-FT300 | 28.9 | 34.9 | 37.1 | 33.2 | 32.3 |
| VLM-FT800 | 45.1 | 44.8 | 54.6 | 45.5 | 47.9 |

Table 3: Performance comparison of different numbers of generated trajectories data. VLM-FT300 refers to the model fine-tuned with 300 successful trajectories, and VLM-FT800 indicates fine-tuning with 800 successful trajectories. We evaluate using a static environment and its corresponding metrics, with 75 tasks in each subset.

## 6 ANALYSIS

### 6.1 INTEGRATION WITH AUTONOMOUS REFINEMENT

As mentioned earlier, our method can also be integrated with an AR approach. Specifically, we continue to guide the VLM agent with the reward model at each action step, and then use GPT-4o as an evaluator to provide reflective feedback. We apply up to three rounds of this integration to assess the performance of both the AR and the integrated methods.

As shown in Figure 2, the performance of the AR method significantly improves from the first to the third round, though the incremental benefit decreases with each additional round. However, the performance curve of the integrated method with the process reward consistently remains above that of the AR (DP+AR) alone, indicating that our approach consistently enhances the AR method's effectiveness.

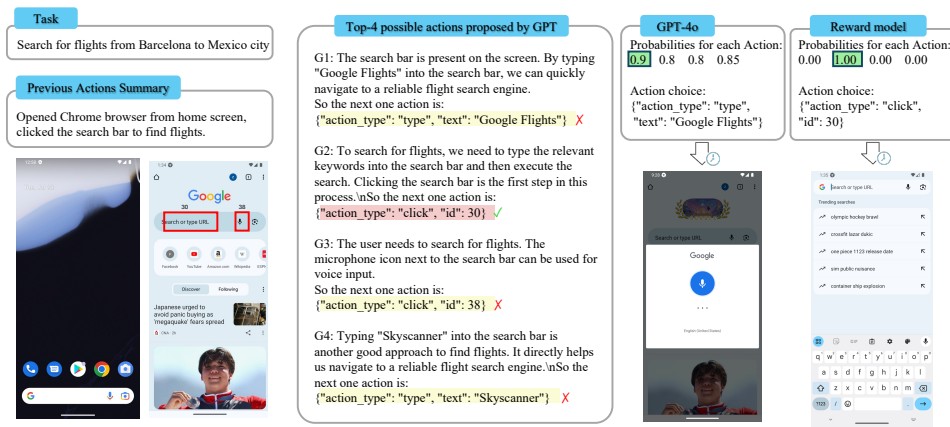

Figure 3: Example of case study. Search for flight in Google Chrome.

## 6.2 COMPARISON OF COMPUTATIONAL EFFICIENCY

We compare the efficiency of several methods across different metrics. Table 4 presents the average token consumption, API cost, and average number of interaction turn required to complete a task for each method. First, the DP and TopK methods show noticeable differences in token consumption and cost. DP consumes 41.0k tokens with a cost of \$0.13, while TopK consumes 59.4k tokens, resulting in a higher cost of \$0.23. Additionally, the number of interaction turns is lower for DP (8.7 turns) compared to TopK (10.6 turns). The AR (n=1) and Integration (n=1) methods consume more tokens, 102.1k and 108.6k respectively, resulting in higher costs of \$0.40 and \$0.41. However, AR (n=2) consumes the most tokens, 129.4k, leading to the highest cost of \$0.51, showing a significant increase in cost with the number of evaluation rounds. These results suggest that the DP and GuidNav methods are more cost-effective, especially for tasks requiring more interaction turn. Although the AR methods consume more tokens and incur higher cost, they may offer different advantages, such as improved accuracy or other performance aspects, depending on the specific task requirements.

| Method | Tokens | Cost | Turn |
|---|---|---|---|
| DP | 41.0k | 0.13 | 8.7 |
| TopK | 59.4k | 0.23 | 10.6 |
| GuidNav | 53.4k | 0.20 | 9.8 |
| AR(n=1) | 102.1k | 0.40 | 17.8 |
| AR(n=2) | 129.4k | 0.51 | 22.5 |
| Integration(n=1) | 108.6k | 0.41 | 19.8 |

Table 4: Efficiency measurement for different methods is analyzed across several metrics. "Tokens" refers to the average token consumption per task. The "Cost" metric corresponds to the average API cost per task, based on the latest GPT-4o pricing (\$5.00 per 1M tokens). "Turn" indicates the average number of interaction turn. The variable $n$ denotes the number of evaluation rounds.

## 6.3 CASE STUDY

Our method demonstrates a superior capability to select the most appropriate actions for complex tasks. This is particularly evident in tasks requiring precise operation sequences. As illustrated in Figure 3, when searching for flights in Google Chrome, the correct procedure involves first clicking the search bar before typing the search content, as highlighted in the previous action summary. Our method correctly identifies this action sequence, ensuring a more accurate and efficient interaction with the environment. In contrast, GPT-4o suggests typing the search content without clicking the search bar first, leading to an incorrect operation.

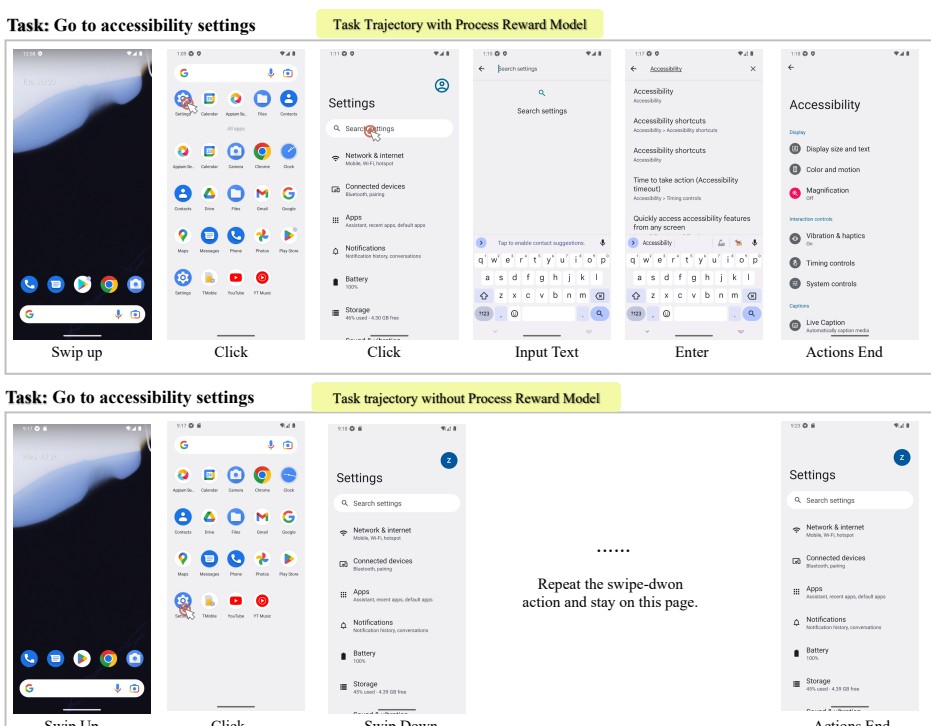

Figure 4: Example of case study. Access the accessibility settings.

Moreover, our approach surpasses other methods in efficiency. The reward model we employ actively learns to prioritize actions that drive progress towards the task goal with minimal redundancy. This is evident in how our method directly focuses on the relevant actions, avoiding unnecessary exploration. For instance, in the bottom scenario of Figure 4, when accessing the accessibility settings, other methods, such as the VLM, exhibit a lack of precision. The VLM continues to swipe down and stucks on this page, reflecting a blind, exhaustive search approach. In contrast, our method identifies the correct path swiftly, avoiding redundant actions and demonstrating a more intelligent task-solving strategy.

This efficiency is not just about completing tasks faster but also about making decisions that align closely with the goal of task, resulting in more robust and reliable performance in dynamic environments.

## 7    CONCLUSION

In this work, we presented the approach to guiding VLMs with a process reward model for improved performance in GUI interaction tasks. Our method addresses the limitations of existing frameworks by enabling VLM agents to optimize actions at each inference step, significantly enhancing action accuracy and task success rates in both static and dynamic environments. Specifically, we demonstrate a near 5% improvement in action accuracy in static GUI environments and a around 15% increase in task success rate in dynamic settings. These results highlight the effectiveness of our process reward model guidance strategy in overcoming challenges such as delayed feedback and local optimization. Furthermore, by incorporating trajectory reflection and retry mechanisms, we further demonstrate advancements in the robustness and efficiency of VLM agents in complex GUI navigation tasks. For future work, GuidNav's generalization can be tested in broader scenarios beyond specific apps, using new benchmarks that cover tasks across operating systems, professional tools, and workflows. Expanding evaluation in these areas will offer deeper insights into its effectiveness in more diverse, real-world contexts.

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

# 8 APPENDIX

## 8.1 REWARD ANNOTATION

To collect training data for the reward model, we utilize the AitW dataset (Rawles et al., 2024b) and employ GPT-4o as the policy model in a static environment for self-play. We automatically label each action at the step level by evaluating its effectiveness. Since the AitW ground truth is represented as coordinates, while our model outputs numeric labels for detected elements, we treat any point within the detected bounding elements as equivalent. According to our rule, a predicted action from GPT-4o is considered correct if both the action type and gesture match the annotator's ground truth actions. Specifically, our evaluation metrics are based on those used in AitW.

- **For click actions**: The action is considered correct if the chosen element by action is within 14% of the screen distance from the ground truth coordinate, or if both the ground truth coordinate and the element selected by generated action fall within the same detected bounding box (expanded to 240% of its original size for action matching).

- **For scroll actions**: the predicted action is considered correct if the scroll direction (up, down, left, or right) matches the ground truth direction.

- **For other actions**: For other actions: The predicted action is considered correct if the action type matches the ground truth. However, for the typing action, both the action type and the typed content must match the ground truth.

## 8.2 SoM SETTINGS

SoM utilizes off-the-shelf interactive segmentation models, such as SEEM (Zou et al., 2024) or SAM (Kirillov et al., 2023), to partition an image into regions of varying granularity. Each region is annotated with marks like alphanumeric labels, masks, or bounding boxes. This enhances the VLM's ability to interpret and understand elements within the image.

In our implementation, we maintain the original SoM configuration (input an image, output bounding boxes and corresponding labels, and overlay the labels onto the image). We choose SAM as the segmentation model and, for each identified entity, assign a unique numeric label positioned at the center. Additionally, we store the labeled screenshot along with the coordinates of each labeled entity for subsequent interactions.

Given the complex structure of GUI interfaces, which often include numerous small entities with relationships such as containment and overlap, we implement specific strategies to ensure accurate interpretation by the VLM. For entities in containment relationships, we retain the identifiers of both the containing and contained entities. In cases of overlapping entities, we prioritize the identifier of the smaller entity to ensure clarity and precision.

## 8.3 PROMPTS

| **Instruction:** |
| --- |
| Provide a summary of the previous actions as follows: {previous_text} , the current thinking steps and the action to be executed as follows: {text}, and the screenshot of the interface after the action is executed. Please summarize the actions above and the status after the action is executed into the new previous actions using descriptive languages brief as possible.(do not speculate on the next move )
Summary: |

Table 5: Prompt for generating historical summarization

| |
|---|
| **Task:** Goal of task |
| **Task Requirements:** |
| Above are two screenshots of a android phone. one is the original screen and the other one has blocks with numeric IDs. You are an AI assistant with a deep understanding of these screenshot and the android phone operations. 
 For example, The home page does not display all installed apps, scrolling up on the home page can open the App Drawer where all the installed apps [If an app is not in the app drawer, it is not installed.] are stored and organized, or you can check whether an app is installed in Google Shop. You need to generate an action based on the current situation, which will be executed automatically without user intervention. 
 The user will not interfere with the entire operation process, such as voice input, which will be regarded as an incorrect operation. 
 Attention: When the user can find the answer from the current page (without needing detailed information), the task can be considered complete." |
| **Available Actions:** |
| available_actions |
| **Summary of previous actions:** |
| *Previous actions*: previous_actions |
| **Instruction:** |
| Based on the above information and the following instruction. please provide your *k* best thought processes (think step by step) and answers for the next one action(only one action),then provide the probability (0.0 to 1.0) that each action contributes to completing the user's requirement at the current stage (according to the image). 
 Answer format for example: 
 G1: <the step-by-step explanation of your thought process (No more than three sentences)> So the next one action is:{"action_type": <action type in <Available Actions>, <the rest information of the action>} 
 P1: <the probability between 0.0 and 1.0 that G1 is correct, without any extra commentary whatsoever; just the probability!> 
 ... 
 Gk: <the step-by-step explanation of your thought process (No more than three sentences)> So the next one action is:{"action_type": <action type in <Available Actions>, <the rest information of the action>} 
 Pk: <the probability between 0.0 and 1.0 that Gk is correct, without any extra commentary whatsoever; just the probability!> |

Table 6: Prompt for generating $k$ possible actions

## 8.4 ACTION SPACE OF AITW DATASET

The AitW dataset consists of a set of predefined actions that the VLM agent can perform in Android environment GUI navigation tasks. The actions are represented as follows:

- `"click"`: Perform a click action on a UI element with a specific `id`. Example: `{"action_type": "click", "id": <numeric IDs on the screen>}`.
- `"type"`: Input text into a UI element. Example: `{"action_type": "type", "text": <text>}`.
- `"navigate_home"`: Navigate back to the home screen. Example: `{"action_type": "navigate_home"}`.
- `"navigate_back"`: Navigate to the previous screen. Example: `{"action_type": "navigate_back"}`.
- `"enter"`: Confirm the current action, typically mimicking an 'enter' key press. Example: `{"action_type": "enter"}`.
- `"scroll"`: Scroll in a specified direction, where the direction can be `"up"`, `"down"`, `"left"`, or `"right"`. Example: `{"action_type": "scroll", "direction": "up"}`.
- `"task_complete"`: Mark the task as completed. Example: `{"action_type": "task_complete"}`.

