# OpenReview forum: "Guiding VLM Agents with Process Rewards at Inference Time for GUI Navigation"
_ICLR.cc/2025/Conference — Submitted to ICLR 2025_

### Official Review · Reviewer_hchw · 2024-10-28

**Soundness:** 2
**Presentation:** 2
**Contribution:** 2
**Rating:** 3
**Confidence:** 3

**Summary:**

The paper proposes leveraging inference time computation to improve action selection for various GUI Navigation tasks. The proposed method includes a process-reward model to validate intermediate steps to help improve success rate on multi-step tasks of GUI Navigation. They also incorporate a reflect and retry strategy to further boost chances of success.

**Strengths:**

- The paper is generally well structured  and easy to follow but is missing some details (see clarification questions below).
- The experiments support the claim that process rewards are beneficial for action selection for sampling at inference time.

**Weaknesses:**

- The paper does not contextualize its contributions correctly by failing to include a discussion on Process Reward Models~(PRM) that have already been proposed and leveraged in multi-step reasoning tasks [References 1-5]. The techniques proposed appear to be an application of these findings on the GUI Navigation domain, while I’m personally of the opinion that applications of techniques on new domains can still present novel insights beneficial to the community, I think the paper in its current state does not do that by glossing over specific details (see Q1-5) and not providing enough domain-specific discussion. I feel the paper will benefit from revealing more domain specific details: What are the nature of these tasks selected for testing the method? How long are typical interactions in dynamic tasks selected? How does the method fare on different tasks – are there specific characteristics of tasks where the current models perform poorly?

- The contribution statement (Section 7) claims more robustness and efficiency in complex GUI navigation but these notions are not defined in the paper anywhere. Improved success rate does not imply efficiency or robustness but these can be interesting to demonstrate by presenting alternate metrics such as the number of steps to goal (efficiency), repeated successful completion under disturbances such as ads in shopping tasks (robustness).

_References_:

[1] Lightman, H., Kosaraju, V., Burda, Y., Edwards, H., Baker, B., Lee, T., Leike, J., Schulman, J., Sutskever, I. and Cobbe, K., 2023. Let's verify step by step. arXiv preprint arXiv:2305.20050.

[2] Li, Y., Lin, Z., Zhang, S., Fu, Q., Chen, B., Lou, J.G. and Chen, W., 2023, July. Making language models better reasoners with step-aware verifier. In Proceedings of the 61st Annual Meeting of the Association for Computational Linguistics (Volume 1: Long Papers) (pp. 5315-5333).

[3] Ma, Q., Zhou, H., Liu, T., Yuan, J., Liu, P., You, Y. and Yang, H., 2023. Let's reward step by step: Step-Level reward model as the Navigators for Reasoning. arXiv preprint arXiv:2310.10080.

[4] Wang, P., Li, L., Shao, Z., Xu, R., Dai, D., Li, Y., Chen, D., Wu, Y. and Sui, Z., 2024, August. Math-shepherd: Verify and reinforce LLMs step-by-step without human annotations. In Proceedings of the 62nd Annual Meeting of the Association for Computational Linguistics (Volume 1: Long Papers) (pp. 9426-9439).

[5] Snell, C., Lee, J., Xu, K. and Kumar, A., 2024. Scaling LLM test-time compute optimally can be more effective than scaling model parameters. arXiv preprint arXiv:2408.03314.

**Questions:**

(Q1) Reward model training (Section 3.1): The annotation process provided in Appendix 8.1 is unclear for self-play trajectories – Is the agent initialized at specific contexts ($S_{t-1}, A_{t-1}, s_t$) in the AitW dataset to then score different actions based on ground-truth action? (Doesn’t this mean we just have more labels for the correct action at each context but not at arbitrary contexts).  If not, how are you effectively mapping arbitrary contexts an agent might reach from the reset distribution of a task to the ground-truth labels in a static dataset? Can you provide more details for how much self-play augmentation was performed for training the process reward models considered in the experiment.

(Q2)  History compression (Section 3.1): How much of the state interactions were compressed for evaluation? Do you compress the states at every decision making step or do you reuse previous compressed history? Am I right in assuming that the history compression is zero-shot GPT-4o (Appendix 8.3 points to a prompt but does not tell which model does this).
- While the prompt explicitly states do not speculate future actions, Figure 3 of the paper indicates that history includes “clicked the search bar to find flights”, which the Top-1 answer seems to assume to be the case (and interestingly ignores the fact that no keyword is visible in the screenshot  for it to type onto).
- Can you comment on the general quality of zero-shot history summarization and the typical lengths of the states compressed by a VLM?

(Q3)  Reflection Thought Generation (Section 3.3): What is the prompt that generates the reflection though for retry? Does it use the full uncompressed context in dynamic tasks? I think key domain specific details for these elements are missing in the paper.

(Q4) On TopK baseline (4.1): The paper states: “We simplify the process by selecting the most probable one. This ensures that the  model still considers multiple possibilities but prioritizes highest probability action for execution” – This statement does not make sense to me, can the authors describe how sampling multiple actions enables this baseline to consider multiple possibilities?

(Q5) For Table 1. Could you comment on the stochasticity of action recommendation of the GPT-4o policy – should the evaluation be performed over multiple seeds for actual Top-K style approaches (like the proposed GuideNav)? In Table 3. Is VLM-FT300 row the GuidNav method that is fine-tuned with 300 tasks from Table 1 – if so, these do suggest some variance in performance for static assessment.

---

### Official Review · Reviewer_Fb3V · 2024-11-03

**Soundness:** 2
**Presentation:** 3
**Contribution:** 2
**Rating:** 5
**Confidence:** 4

**Summary:**

The approach guides VLM agents with process supervision by a reward model during GUI navigation and control at inference time.  The agent can optimize its actions at each inference step.  Significant performance gains are presented in action accuracy  in static environments and 15% increase in task success in dynamic environments.  Trajectory reflection and retry increases task success.

Section 1 describes the introduction, along with the argument that VLMs are too resource intensive.  DigiRL, which combines advantage weighted regression with online reinforcement learning has high computational and time costs and a complex training process, with lots of online interaction data.  These are addressed by the proposed approach, GuidNav, which has a process reward model during interaction inference for GUI navigation and control tasks.  A process reward  model is learned based on a limited amount of human demonstrations and synthesized data generated by the VLM.  AitW is used to evaluate in both dynamic and static settings.  Section 2 describes related work.

Section 3 describes the approach. Training data for the reward model comes from a) human demonstrations and 2) self-play with VLMs.  History is summarized using the VLM in text and is input to the reward model.  The training objective is to minimize the difference between the predicted and actual rewards.  In section 3.2, the authors again summarize the previous actions and states and then generates k possible actions.  The “VLM” selects the action with the highest reward.  The VLM will then reflect by generating a “reflection thought”, which is incorporated into the prompt of the next iteration.

Section 4 describes various baselines, the datasets.  GPT4-o was used for the policy model and CogVLM2 was used for the process reward model.  2 annotators are used to measure final task success.

**Strengths:**

Overall, the paper is in an interesting area and the authors have performed substantial experiments.  That said, I think there are still many questions to be answered, and, as written, the paper would not be reproducible due to lack of details about training the process reward model (the most important part of the paper).  Timing numbers (e.g., in s) should also be added to Table 4.  See below some questions that would help clarify the paper.

**Weaknesses:**

see above

**Questions:**

Some major questions include:
* How good is the process reward?  Did you evaluate the process reward model in the context of per-step reward prediction?
* Even though you call it out in the introduction, I’m not sure why it’s not on the table to update VLM’s weights as you explore the environment?
* Summarizing state and actions is an interesting way to condition the reward model.  How good are the summarizations?
* I don’t understand the choice of GPT-4o and CogVLM2. Why not try to understand the impact of VLM choice on the downstream performance?  Like you could have at least picked a few state-of the art models and compared them (in addition to a closed source one).
* Nowhere in the main paper do you describe training the process reward model (e.g., what model, hyper parameters, training setup).
* In Table 4, actual inference time should be added here.

P2.  “Can also been” -> “can also be”
P9. “Swipe-dwon” -> “swipe-down”
P9. “Down and stucks” -> “gets stuck”

---

### Official Review · Reviewer_MSFb · 2024-11-04

**Soundness:** 3
**Presentation:** 3
**Contribution:** 2
**Rating:** 5
**Confidence:** 3

**Summary:**

This paper develops an approach for a foundation model to be able to execute users' language commands for navigating a graphic user interface (GUI). The approach consists of three steps. First, the system trains a reward model given human demonstrations to predict the reward (a measure of alignment between a human trajectory and a candidate trajectory) of a trajectory as a sequence of actions towards accomplishing the user's command. Second, a VLM generates a set of candidate actions, the reward modeling component assesses those actions, and the best is executed in the environment. After some number of steps, a model is queried to determine if the plan succeeded; if yes -- done; if not, then the system incorporates feedback from that failed attempt towards another attempt.

**Strengths:**

- The notation and formulation of Equations 1 and 2 are helpful and align with a POMDP formalism nicely.

- Figure 2 shows positive results improving the success rate of the methods as a function of the number of refinement steps.

- The paper is mostly well-written and clear and most details are present to support reproducibility.

- Table 4 is a helpful addition to explore computational efficiency considerations.

- While simple, the case study in Section 6.3 is helpful.

**Weaknesses:**

-"a around" should be "an around"

-The grammar here could be improved as the phrasing is awkward: "Apart from agent framework"

-"VLM - generated" should be "VLM-generated" in Line 173

- If a trajectory fails to execute the desired command and the system tries to start over, the initial state, $s_1$, may not be the same if the system is trialing the trajectories in the real world. The paper does not seem to address the problem of undoing or resetting the system back to its initial state during the reflection and retry component. If the system used a world model, then this would not be a problem, but it doesn't seem that a world model is discussed or used.

- It would be helpful for the x-axis of Figure 2 to not have increments of 1/2 as the actual step size is 1.

- The paper could be improved by including an analysis to show how often failures in each iteration lead to more problems at subsequent iterations as opposed to getting to reset to the initial state.

- The results could be improved if there were confidence intervals on the results in the tables and some notion of temperature and how that hyperparameter might effect results.

- Table 3 is helpful to show the performance of the system improves with a 2x-3x increase in training data size; however, it would be even better if a dimension was included for the quality of the data.

- It would have been helpful to include additional benchmarking tasks outside of AitW.

- An additional analysis showing the performance of the system as a function of task complexity (e.g., number of steps required in the optimal plan) would have been useful.

- The approach is a relatively straight-forward combination of

- It would be helpful to add details about whether the reward model and the VLM policy are the same system. If so, how is the VLM trained to output the reward? Is this just intrinsic or through prompt engineering or is the VLM literally modified to add a scalar output head and trained from scratch or fine-tuned?

- The approach seems to combine some relatively simple elements to create a new system; the paper could be improved by being more clear about the novelty and intellectual merit of the approach more than just combining simple elements of prior work.

**Questions:**

- Why is Equation (3) written like a POMDP where $\gamma = 0$? Why not start addressing the credit assignment problem and sequential planning more explicitly through a Q-function?

- Why perform greedy action selection (Equation 6) instead of doing some kind of Monte Carlo rollouts and a Bellman-based backup?

- Can the authors please note which of the baselines in Tables 1 and 2 use an iterative mechanism for "reflection" such that they "learn" from prior trials like the proposed method does?

- How are the dynamic vs. static settings different than in-distribution vs. out-of-distribution evaluation?

- How does system performance degrade as the expert training data's quality (efficiency) degrades?

- How high does the performance need to be before a user adopts the system?

- For the various components of the system, which are actually learned via gradient descent vs. through in-context learning? Which components are a part of the same VLM vs. separate models?

---

### Meta-Review · Area_Chair_jrLT · 2024-12-19

**Metareview:**

(a) The paper introduces GuidNav, a method that uses a process reward model to guide Visual Language Model (VLM) agents during GUI navigation tasks. The core claim is that by providing step-level rewards during inference, the VLM agent can optimize actions at each step, thus improving overall performance in both static and dynamic GUI environments. GuidNav achieves a 5% improvement in action accuracy in static environments and a near 15% increase in task success rates in dynamic environments.

(b) Strengths:
- Novel approach: The paper introduces a novel approach using a process reward model to guide VLM agents during GUI navigation.
- Performance gains: The experimental results show significant improvements in action accuracy and task success rates in both static and dynamic environments.

(c) Weaknesses:
- Lack of clarity on novelty: Some reviewers felt the approach was a relatively straightforward combination of existing elements and lacked sufficient novelty.
- Missing details: The reviewers noted several missing training details, regarding the model and reward model training process
- Lack of evaluation on the process reward model in the context of per-step reward prediction.
- Missing experiment design details.
- Unclear claims: The paper makes claims about robustness and efficiency, but these concepts were not defined and not substantiated by any evidence.

(d) My decision is to reject. As the reviewers pointed out, the main reasons are lack of novelty and context, missing experiment details, limited evaluation, unclear claims, and methodological concerns about the experimental design (e.g. the appropriateness of the baselines). The paper did present a promising approach, and demonstrated some positive results, but the shortcomings and concerns outweighed the strengths, making it unsuitable for acceptance in its current state.

**Additional Comments On Reviewer Discussion:**

The authors did not respond to the reviewers.

---

### Decision · Program_Chairs · 2025-01-22

Reject